# ADAM10 Site-Dependent Biology: Keeping Control of a Pervasive Protease

**DOI:** 10.3390/ijms22094969

**Published:** 2021-05-07

**Authors:** Francesca Tosetti, Massimo Alessio, Alessandro Poggi, Maria Raffaella Zocchi

**Affiliations:** 1Molecular Oncology and Angiogenesis Unit, IRCCS Ospedale Policlinico S. Martino Largo R. Benzi 10, 16132 Genoa, Italy; alessandro.poggi@hsanmartino.it; 2Proteome Biochemistry, IRCCS San Raffaele Scientific Institute, 20132 Milan, Italy; alessio.massimo@hsr.it; 3Division of Immunology, Transplants and Infectious Diseases, IRCCS San Raffaele Scientific Institute, 20132 Milan, Italy; zocchi.maria@hsr.it

**Keywords:** ADAM, metalloproteinases, subcellular trafficking, vesicles, exosomes, signaling, cancer, ADC, immunomodulation

## Abstract

Enzymes, once considered static molecular machines acting in defined spatial patterns and sites of action, move to different intra- and extracellular locations, changing their function. This topological regulation revealed a close cross-talk between proteases and signaling events involving post-translational modifications, membrane tyrosine kinase receptors and G-protein coupled receptors, motor proteins shuttling cargos in intracellular vesicles, and small-molecule messengers. Here, we highlight recent advances in our knowledge of regulation and function of A Disintegrin And Metalloproteinase (ADAM) endopeptidases at specific subcellular sites, or in multimolecular complexes, with a special focus on ADAM10, and tumor necrosis factor-α convertase (TACE/ADAM17), since these two enzymes belong to the same family, share selected substrates and bioactivity. We will discuss some examples of ADAM10 activity modulated by changing partners and subcellular compartmentalization, with the underlying hypothesis that restraining protease activity by spatial segregation is a complex and powerful regulatory tool.

## 1. Introduction

Increasing evidence indicates that vesicular compartments and intracellular organelles work as structural scaffolds to coordinate specificity and temporal activity of functional hubs in cell signaling, cell death/survival programs and enzymatic functions, including proteolysis. It is not by chance that most enzymes are membrane or membrane-associated proteins [1]. Different forms of subcellular segregation indicate that the dynamic spatial pattern of signaling enzymes and proteases have reciprocal, bidirectional functional implications [2].

The evidence that proteases, the most abundant class of enzymes, are able to work in unpredicted environments was first recognized for matrix metalloproteinases in their cell-associated or soluble forms, reviewed by Shimoda and Khokha [3]. The activity of cell membrane-associated proteases, in particular, was mostly deduced from their role in cleavage and/or activation of membrane substrates, such as receptors’ extracellular domain, adhesion and migration molecules, or immune modulators.

Here, we focus on ADAM10, a member of the endopeptidase family, to highlight recent advances in the knowledge of regulation and function of these enzymes at specific subcellular sites, or in multimolecular complexes. ADAM17 will be considered for comparison, since the two enzymes belong to the same family of metalloproteases, and share selected substrates and biological activity [4].

Our overview will converge on recent and early evidence of a “social” role of ADAMs, in particular ADAM10, looking at its “relationships” with contextual membrane-associated partners. Then, we will consider signals from the extracellular microenvironment and the possible negative regulation of spatially segregated ADAMs. Finally, the emerging evidence for intracellular ADAM10 function, and for the extracellular transfer of this enzymatic activity, will be presented.

## 2. ADAM10 Processing, Maturation and Trafficking

### 2.1. Structural and Spatial Determinants of ADAM10 Function

ADAM endopeptidases, in particular ADAM10 and ADAM17, are the primary enzymes responsible for ectodomain shedding of membrane proteins in eukaryotes, with a set of preferentially overlapping substrates and specific targets [5,6]. For example, tumor necrosis factor α (TNFα), cleaved by ADAM17 (also named TNFα convertase or TACE) and ADAM10, is involved in the development of some human tumors where the two enzymes are overexpressed [7]. For these reasons, ADAM10 and ADAM17 have been proposed as biomarkers and possible targets of therapy in cancer [8].

Proteolytic shedding is a restricted cleavage close to the transmembrane domain of integral proteins, producing an ectodomain fragment, which is secreted in a free or vesicle-associated form in some cases, and a membrane-associated fragment. ADAMs are single pass type I membrane glycoproteins belonging to the metzincin superfamily, which consists of five families of metalloproteases (astacins, adamalysins, serralysins, matrix metalloproteases, MMP, or matrixins, and pappalysins) [9]. Several recent comprehensive reviews covered almost all aspects of ADAMs biology [4,10,11,12].

The paradigmatic view of ADAM10 and ADAM17 topology is that of cell membrane-anchored enzymes working in the pericellular space as mature α-secretases derived by catalytically inactive precursors. This evidence refers especially to amyloid precursor protein (APP) cleavage by ADAM10 [13]. However, recent insights have revealed sequence and structure determinants indicative of complex intracellular trafficking and sites of action (reviewed in [14]). Curiously, ADAM10 is described as a protease resident in the Golgi apparatus and derived vesicles (i.e., as annotated in UniProtKB, O14672 ADA10_HUMAN; https://www.uniprot.org/uniprot/O14672 accessed 10 February 2021), despite studies on cell membrane targets contributed major insights into its enzymatic activity.

Interestingly, only about half of the ADAM family members are active proteases, including ADAM10 and ADAM17, while the remaining eight members (i.e., ADAM2, 7, 11, 18, 22, 23, 29, 32) are predicted to be catalytically inactive and are involved in other biological functions, such as protein–protein interaction, axon growth and neurogenesis, and infection (reviewed in [15,16]).

ADAM10 modular architecture shows hallmarks of most MMPs (Figure 1). It includes an extracellular domain with a N-terminal signal sequence, a prodomain (PD), the largest metalloproteinase domain (M) containing one zinc ion coordinated within the zinc-binding consensus motif (HEXGHXXGXXHD) of the catalytic site, an autoregulatory disintegrin domain (D), and a cysteine-rich (C) domain, able to suppress the metalloproteinase activity. The PD is removed to give the active mature form by proprotein convertases (PC)7 and furin (RXXR consensus sequence) (see Section 2.3). A transmembrane (TM) domain anchors ADAMs to the cell, and a short cytoplasmic tail containing a SH3 binding domain shows a regulatory role on ADAM10 activity [10].

ADAM10 ectodomain structure revealed the possibility of an autoinhibition of the active site by the C-terminal tail of the adjacent ADAM10 protomer (the smallest functional unit), mimicking the substrate, which could prevent uncontrolled hydrolysis of target proteins (Figure 1). Remarkably, two pairs of ADAM10 protomers are apparently needed for this cooperative regulation to occur [17]. In turn, ADAM dimerization can be inhibited by TIMP3, the MMP inhibitor with the broadest spectrum of substrates, limiting the enzymatic function [18,19].

ADAM10 homodimerization has been reported to depend on the transmembrane domain keeping the cytoplasmic tail close to the membrane surface [18]. In this way the ADAM10 cytoplasmic domain, which contains an intrinsically disordered proline-rich region and polar residues favoring protein aggregation, acquires an ordered structure [20,21,22]. It is tempting to speculate that this mechanism, which can in principle produce more than a single ordered structure [23], might contribute to enzyme specificity by establishing multiple interactions of ADAMs with different partners.

### 2.2. Regulation of ADAM10 Trafficking and Export

Two classes of molecules facilitate trafficking and segregation of ADAMs in different cellular compartments: tetraspanins and rhomboids.

Tetraspanins (Tspan) are cell-surface proteins that span the membrane four times, forming two extracellular loops. The “tetraspanin web” [24] facilitates protein progression and lateral mobility of partners along the intracellular vesicular network, accumulation in storage depots, or clustering at the plasma membrane. The subgroups of TspanC8, having eight cysteine residues in the extracellular domain, regulate ADAM10 export from the ER, progression and maturation along the secretory pathway [10,14,24,25,26,27]. Association with members of theTspanC8 family (Tspan5, 10, 14, 15, 17, 33) is mediated by the ADAM10 disintegrin cysteine-rich regions (Figure 1). Remarkably different Tspan/ADAM10 complexes can cleave defined substrates [26]. It has been proposed that ADAM10 can adopt distinct conformations in association with different TspanC8 [27]. Furthermore, ADAM10 stimulation by specific chemical inducers has been shown to increase processing of specific substrates with the aid of TspanC8. This TspanC8-mediated mechanism appears to be another cell-specific modality for switching on ADAM10 activity against selected targets. For example, transfection with Tspan15 was shown to strongly reduce ADAM10-mediated Notch and APP cleavage, while stimulating N-cadherin and CD44 processing. Overexpression of Tspan5 instead impaired N-cadherin processing, with no apparent effect on APP [26].

Among TspanC8, Tspan15 has been established to assist ADAM10 exit from the ER (Figure 1) and further processing [26], as shown by the molecular interaction between the two proteins in mammalian cell lines [28,29]. An arginine-rich ER retention motif, limiting ADAM10 trafficking and inhibiting its sheddase activity, has been mapped in the cytoplasmic C-terminal domain of ADAM10 [30,31], thus indicating the existence of a potential ER-resident (possibly inactive) form and another level of regulation. Facilitation of ADAM10 export from the Golgi apparatus has been shown to depend on PKC phosphorylation of synapse-associated protein 97 (SAP97) SH3 domain (Figure 1), which interacts with the ADAM10 SH3 binding domain [32].

In the case of ADAM17, intracellular trafficking, lateral mobility and membrane location is governed by proteolytically inactive iRhom1 and iRhom2 rhomboid proteins, instead of Tspans, and iRhom2 also assists ADAM17 maturation by promoting its exit from the ER [33]. There is also evidence that different TspanC8s and iRhoms are not only responsible for intracellular transport, but also target the ADAMs to distinct substrates [1,26,27,28,31,33,34,35,36,37]. Thus, ADAM10 can behave as more than one molecular scissor, depending on the Tspan involved, while the same can be thought for ADAM17 on the basis of the iRhom recruited. These specific binary interactions seem to be regulated by different regions of the molecules. Indeed, TspanC8 and ADAM10 interaction is mediated by their extracellular region, largely involving the cysteine-rich domains [27], while the interaction of ADAM17 with rhomboids is mainly mediated by its cytoplasmic tail [35]. This is not surprising, since ADAM10 and ADAM17 share only 30% of amino acids identity in human. For example, the cysteine-rich domain in ADAM17 is much shorter than in ADAM10 (38 vs. 118 amino acids, respectively), and thus it might not be suitable for the interaction with TspanC8.

The dynamics, regulation and biological implications of ADAMs export and progression along the secretory compartment to the cell surface, are recent acquisitions. In particular, the details of ADAM10 activation as a function of its structural determinants, prodomain removal and subcellular localization in the secretory pathway are a topic of intense ongoing research, in light of the recent evidence of ADAM10 biological role in tumors and neurodegenerative diseases such as Alzheimer’s disease [10]. In turn, ADAM17 inhibition was previously shown to relay on additional structural levels of regulation, apart from prodomain removal [38]. Notably, recent evidence has revealed the existence of circulating auto-antibodies directed against the ADAM10 prodomain in colorectal cancer (CRC) patients [39]. Similarly to what was observed in LoVo CRC cells, accumulation of unprocessed ADAM10 could be due to defective furin proprotein convertase or altered expression of Tspan [40]. The questions of how auto-antibodies against unprocessed ADAM10 are produced, and whether unprocessed ADAM10 in CRC patients is released in a soluble or vesiculated form, remain unanswered issues. These molecular details could be relevant for the accessibility of both intracellular and vesicular ADAM10 to specific inhibitors.

### 2.3. ADAM10 Activation and Proteolytic Ensemble

Cleavage by furin and PC7 of the PD, the first known mechanism of ADAM10 activation (Figure 2), has been shown to be dispensable for ADAM10 activation in certain experimental conditions. Indeed, the rapid stimulation of ADAM10 due to extracellular calcium influx is not significantly affected by proprotein convertase inhibitors, making a predominant role of prodomain removal unlikely [31]. An interesting clue on the role of the immature, apparently nonfunctional ADAM10 form is the discovery of autoantibodies against ADAM10 prodomain in CRC patients, correlating with a significant extension of recurrence-free survival median time [39]. On the other hand, the existence of a tetrabasic cleavage site on ADAM10 indicated that proteolytic activation by proprotein convertases PC7 or furin in the secretory pathway was possible in principle [41,42]. A second ADAM10 mechanism of activation is the regulated intramembrane proteolysis (RIP) (Figure 2).

The discovery of RIP, the cleavage of peptide bonds within the membrane lipid bilayer, gave birth to a round of overturning studies [43,44,45]. Enzymes acting through RIP show distinguished structural features and an intramembrane active site. RIP occurs at the level of the main subcellular compartments involved in metabolic and stress signaling, including the ER, taking part to protein quality control and ER degradation, in the Golgi apparatus devoted to protein maturation, and in endolysosomal vesicles.

RIP engages numerous intramembrane-cleaving proteases (I-CLiPs) (Figure 2), including the membrane-embedded rhomboid and rhomboid-like proteases in humans (RHBDL1-4 and PARL), Rce1-type glutamyl proteases, and aspartyl proteases (signal peptide peptidases and γ-secretases) [44,45]. The human orthologs of rhomboid proteases RHBDL1-4, whose proteolytic activity and specific substrates are under investigation, also show a regulatory function in protein progression from the ER to the Golgi complex and the cell surface [2]. Interestingly, RHBDL2 was demonstrated to cleave epidermal growth factor (EGF) and trigger EGFR signaling in cells where metalloproteases, including ADAM10/ADAM17, were inhibited, suggesting a possible crosstalk among the two classes of proteases in processing RTK substrates [46].

Different ADAM endopeptidases take part in RIP, often in a first, priming phase of protein processing. The two-step cleavage of RTK ectodomain within the extracellular juxtamembrane domain (ECD) is operated by members of the ADAM family (ADAM10 and ADAM17) working as sheddases, which cleave at a distance of about 30 amino acids from insertion into the cell membrane [47]. This allows access to the γ-secretases that subsequently cleave the cytosolic portion of the transmembrane domain (Figure 2).

Activation of Notch, a prominent ADAM10 target in particular, is an exemplary multiproteolytic process involving RIP, furin, membrane type 1 matrix metalloproteinase (MT1-MMP) and ADAM10 [48,49] (Figure 1). ADAM10 and Notch must be present in the same cell to interact, to allow the proteolytic cleavage in this case [34], although this does not occur for all ADAM10 substrates.

An alternative pathway of unconventional protein secretion for ADAM17 that can bypass iRhom-dependent ER to Golgi transfer, normal glycosylation and prodomain removal has been described, favoring ADAM17 insertion into extracellular vesicles [50]. These facts, as well as the evidence of overexpression of ADAM10 immature form in CRC, provide further insights into ADAM10 quality control governing ADAM10 trafficking along the secretory pathway with respect to ADAM10 dysregulation in cancer and in tissue homeostasis in general.

## 3. ADAM10 as a Bridge between Extracellular and Intracellular Environments

### 3.1. ADAM10 Activity and Intracellular Signalling

The link between membrane or intracellular trafficking, signal transduction and substrate proteolysis is well documented for receptor tyrosine kinases (RTK) anchored to the plasma or vesicular membranes. Cleavage cascades through RIP generate RTK carboxy-terminal fragments that undergo γ-secretase-mediated splitting into intracellular domains (ICD), which gain autonomous function after restricted association with signaling organelles (nucleus, mitochondria) [47,51]. Notably, ADAM10 and ADAM17 are involved in a rate-limiting proteolytic event preliminary to RIP [52]. GTPases anchoring to subcellular organelles, too, regulate their input function on signaling frameworks [53] (Figure 2).

At least half of 55 human RTKs apparently function as substrates for γ-secretases and RIP [43,54], linking regulation of RTK-triggered signaling to preliminary proteolytic events. Other examples include signaling by G-protein coupled receptors (GPCR) and ADAMs activity, which are interconnected processes [55]. Signaling enzymes such as Rab GTPases as well regulate intracellular membrane trafficking of proteases (Figure 1), conveying messages to modulate protein transit through the ER to the Golgi apparatus and ER-associated degradation [53]. In this context, ADAM10 and ADAM17 trafficking still present unclarified aspects.

For many enzymes, the association with intracellular organelles modulates spatio-temporal biological effects. This is the case for glycogen synthase kinase3 (GSK3) localized in the cytoplasm, the nucleus, in mitochondria and multivesicular bodies (MVB) [56,57]; in fact GSK3, an archetypal moonlighting protein (a molecule that during evolution acquires the ability to perform more than one function) changes the intracellular site of action depending on the metabolic state of the cell [58]. For instance, GSK3 associates with a molecular scaffold which tethers mitochondria to the ER under stressful conditions [59].

Intriguingly, ADAM10 and ADAM17 often directly or indirectly regulate the activity of multiprotein complexes connected with subcellular organelles arranged in highly organized assemblies, which narrow specificity of enzymes and signaling during different stress conditions. ADAM10 and ADAM17, for example, are the main thioredoxin-1 cleavage enzymes generating the truncated thioredoxin-80 form, which activates the inflammasome [60]. The cytoplasmic inflammasome NLRP3, a paradigmatic multimolecular platform with a role in stress and ageing, can in turn recognize organelle-associated molecular structures such as ER engulfing protein aggregates indicative of ER stress [61,62]. The beclin1 interactome is another signaling hub that regulates autophagosome formation for cleansing of dysfunctional mitochondria (mitophagy) and affects the processing of APP, involved in Alzheimer disease, by ADAM10 [63].

### 3.2. Small Signaling Molecules as Metabolic Indicators of ADAM Activity

Recent evidence indicates that ADAM activity changes in response to small signaling molecules, above all calcium (Ca^2+^) [31,64,65], nitric oxide/cGMP, cAMP, and related signaling pathways (i.e., GPCR induced-PKC and other membrane-bound kinases) [13] (Figure 1).

How the inside-out communication of intracellular stimuli is translated into protease activity is a matter of intense investigation for ADAMs [66]. The interaction with surface-exposed phosphatidylserine, following elevation of intracellular Ca^++^, for example, is one of the mechanisms controlling the biological function of ADAM10 (and ADAM17) at the cell membrane [65].

We will illustrate some examples of how small molecules influence ADAMs accessibility to substrates and subcellular location in specific membrane microdomains and organelles. This is in light of the evidence that most of these small molecules operate as messengers in multiprotein complexes requiring physical interactions at membrane level and functional coordination between components. Several data on small molecules regard ADAM17 activity, while evidence of ADAM10 regulation comes from neurobiological studies on AD and APP processing, a topic recently illustrated in dedicated reviews [13,63]. However, this relevant pathophysiological context can open unexplored avenues on ADAM10 biology. For instance, the evidence of physical association of ADAM10 with 5-hydroxytryptamine receptors mediated by the cAMP sensor exchange protein activated by cAMP [67] might encourage the investigation of cAMP on ADAM10 location and activation; indeed, new data highlight the importance of Golgi in cAMP response, other second messengers, GPCR signaling and membrane dynamics [68,69]. The dependence of ADAM10 activity on physiological Ca^2+^ flux oscillations (purinergic receptor agonists), or pharmacological induction by Ca^2+^ ionophores (ionomycin) [31,64,65,70] might be another evidence of ADAM10 need for membrane contact and dynamics to achieve specificity of action. The ADAM10 targets CX3CL1 (fractalkine) and CXCL16 chemokines are unique in their family to occur as soluble as well as type I transmembrane proteins [70,71]. In this system, the use of ionomycin revealed an increased processing of membrane CX3CL1 and CXCL16 by ADAM10. Analysis of the membrane-associated C-terminal cleavage fragments revealed multiple cleavage sites used by ADAM10, one of which preferentially used upon Ca^2+^ influx. In this context, other molecular events influencing ADAMs activity are PKC activation by phorbol esters, such as phorbol-12-myristate-13-acetate (PMA), mimicking diacylglycerol, perturbation of cholesterol metabolism by statins and zaragozic acid, and protein tyrosine phosphatase inhibition by the insulin-mimetic pervanadate [72].

PMA enhances ADAM17-mediated shedding, while ADAM10 is less affected and rather responds to pervanadate (see below). PMA stimulation is then used as a biochemical criterion to distinguish the activities of the two enzymes [70,73]. Of note, PMA diverts mature ADAM17 release into exosomes, while decreasing its surface expression in lung A549 tumor cells [73].

Cholesterol depletion by methyl-beta-cyclodextrin in breast cancer cell lines increases vesicle release and treatment with pervanadate regulates ADAM10 relocation in microvesicles derived by the Golgi apparatus, while PMA favors ADAM10 transport to the cell membrane [74]. In this way, two distinct pathways for cleavage of the adhesion and migration molecule L1 (CD171) can take place [74]. Interestingly, a cholesterol molecule can bind a cavity within the TSPNs transmembrane domains [75] and its removal might result in conformational changes that affect/regulate the interaction with ADAM10 [10].

Pervanadate was demonstrated to inhibit budding of nascent secretory vesicles from the trans Golgi network (TGN) [76], and it is often used to activate ADAM10/17 shedding of substrates, including release of L1 in exosomes stimulated by membrane-bound Src kinase [74]. More data are needed to assess the role of this small molecule in ADAMs sorting.

Taken together, these data confirm spatial relocation of ADAMs, in particular ADAM10, and their substrates, depending on the cell activation state by internal or external stimuli, or metabolic conditions [77]. In this context, the plethora of molecules above mentioned might have a role in allosteric regulation of ADAMs activation at distance. Indeed, conformational perturbation inside a macromolecule (allostery), due to external stimuli (e.g., temperature or allosteric drugs such as maraviroc which targets the chemokine receptor CCR5) can propagate inside the cell through various pathways, as a wave of activation that involves many molecular complexes [78]. In the case of ADAM10, different external stimuli may result in the preferential location of the sheddase in distinct organelles (e.g., MVB rather than exosomes), leading to the selection of certain substrates (e.g., CD30 rather than TNFα) and directing the final biological effect.

## 4. ADAMs Substrates and Partners

### 4.1. Multimolecular Complexes and Subcellular Compartments

ADAM10 stands out among endopeptidases because of its broad specificity in membrane protein ectodomain shedding. Indeed, ADAM10 targets include about 100 substrates [6]. The ADAM-type sheddases are the prominent enzymes involved in the processing of growth factors or cytokines and their receptors (TNFα, FasL, TNFRSF8/CD30), signaling molecules (Notch, ephrins), EGFR ligands (proEGF, heparin-binding HB-EGF, amphiregulin, and betacellulin). Some substrates including for example Notch, TNFα, EGF, CD44, CX3CL1, IL6R, APP, meprin, L1, Delta, Klotho, are shared targets of ADAM10 and ADAM 17, depending on diverse cellular conditions [4,79,80,81,82]. RIP and ADAM10 are responsible for canonical ligand-dependent Notch receptor activation. Indeed, Notch proteins are the most important ADAM10 substrates in development, as demonstrated by the embryonic lethality of ADAM10 knockout mice that phenocopies Notch1/4 knockout mice [48].

Other ADAM substrates are involved in cell-to-cell adhesion (cadherins) or modulate the cellular interactions in the immune system, such as the stress-induced ligands for the natural killer 2D (NKG2D) receptor, including the MHC class I chain-related gene A/B (MICA/B), or the UL16 binding proteins (ULBPs) involved in immunomodulation, to name a few [4,82]. Assembly in multiprotein complexes together with association with vesicular structures is a way cells use to control and specify instructions for enzyme activity. Emerging evidence on ADAMs locations indicates that this kind of control helps substrate discrimination, timing and mode of action also for ADAM proteases.

Clustering of macromolecular complexes is a cell strategy relevant to countless biological functions. Most ADAM10 and ADAM17 substrates are part of multiprotein complexes, imparting different coordinated levels of regulation to proteolytic processing and substrate accessibility (Table 1).

The family of ephrin (Eph) RTK and their ligands are important regulators of immune cell development and immune evasion in the tumor microenvironment [80]. Several lines of evidence obtained in nonimmune models indicate that ADAM10 shedding of Eph requires binding to Eph receptors and scaffold proteins, which create a defined structural recognition module and an upper level of regulation for the proteolytic activity of ADAM10 [81]. An EphB–E-cadherin–ADAM10 complex with glycosylphosphatidylinositol-anchored (GPI) Eph-B ligands is an example of a recognition module for cleavage of E-cadherin by ADAM10, an event which governs sorting of Paneth cells within the crypt stem cell niche [82]. Shedding of Eph by ADAM10 in the juxtamembrane region of their ectodomain has been shown to depend on EphB2 binding to the scaffold protein flotillin-1, a lipid rafts and exosomal marker, which stabilizes this complex, which is critical for the correct morphogenesis of the neural tube [83]. Furthermore, ADAM10 has been demonstrated to be required for EphA1 endocytosis, along with clathrin and dynamin, upon binding to the RTK EphA2 [84].

T cell receptor (TCR) has been shown to regulate ADAM10- and ADAM17-induced shedding of the immune check point molecules Lag3, Tim-3 [85] and PIK3IP1 (transmembrane inhibitor of PI3K or TrIP) [86], a putative transmembrane regulator of PI3K, with distinct mechanisms. To our knowledge, a “sheddase recognition motif” has not yet been identified in protein target sequences. A putative ADAM10/17 target motif was recognized in angiotensin-converting enzyme 2 (ACE2) [87]. ACE2 shedding by ADAM17 revealed another exemplar sophisticated control of ADAM17 activity by neuronal Angiotensin II type 1 receptor (AT_1a_R) through reactive oxygen species and ERK signaling in neurogenic hypertension and the brain renin-angiotensin system [88].

Going through remarkable examples of ADAMs in multiprotein complexes, ADAM10 is considered the Staphilococcus aureus α-toxin receptor. ADAM10, in fact, has been shown to disrupt the integrity of epithelial adherens and tight junctions, which hamper pore-forming bacterial toxins. The TspanC8 member Tspan33 is a major host factor for α-toxin. Tspan33 is able to dock ADAM10 at junctions, where it clusters depending on properly organized actin filaments [89]. Then ADAM10 in complex with pleckstrin homology domain-containing, family A member 7 (PLEKHA7), PDZ-domain containing protein-11 (PDZ11) and afadin, locking ADAM10 to the complex, forms stable α-toxin pores by a dock-and-lock mechanism, enhancing bacterial cytotoxicity [85]. The requirement for an ordered actin network was also observed for ADAM10 positioning at the T cell-antigen presenting cell interface for Notch activation [90].

ADAM10 participates in cell-cell contact in other systems, such as the cochlear sensory epithelium. EphA4 in fact forms a complex with E-cadherin and its sheddase ADAM10, which promotes the destruction of E-cadherin-based adhesions [91].

These examples indicate a propensity for ADAM10 to recognize protein targets in conformational ensembles, elevating the complexity of substrates and interacting proteins from single molecules to protein–protein interfaces with a defined geometry.

### 4.2. Early Clues and Recent Evidence of Active Intracellular ADAM10

After their discovery, early investigations suggested that ADAM location was not restricted to the plasma membrane, and ADAM10 was no exception. Sparse early data reported potential intracellular ADAM10 activity. Millichip et al., in 1998 described a requirement for microsomal membrane localization of ADAM10 from bovine kidney in order to obtain optimal degradation of myelin basic protein (MBP) and collagen IV, revealing a type IV collagenase activity [92]. In contrast, ADAM17 subcellular function was investigated in more detail [33,73,93].

Apart from synaptic vesicles, intracellular ADAM10 often shows a punctuated pattern, indicative of accumulation in small vesicles [94,95]. Several lines of evidence point to an active role and biological significance for endosomal ADAM10, requiring a low pH to differentially cleave substrates. Mathews et al., demonstrated that CD23 processing by ADAM10 occurs in the endosomal compartment, after CD23 endocytosis [96]. Besides CD23 shedding, endosomal ADAM10 was necessary for CD23 sorting into B cell-derived exosomes [96]. ADAM10 also processes CD44 and L1 within endosomal vesicles [94,97], and requires pre-association with these substrates prior to cleavage [66]. Ectodomain shedding by ADAM10 of sortilin, a chaperone that traffics neuronal brain-derived neurotrophic factor (BDNF), was localized at the cell surface and in an intracellular compartment [98].

An in-depth investigation describing active ADAM10 in the lysosomal compartment reported co-localization and processing of the substrate FasL in T cells upon TCR activation [99]. Lee et al., more recently demonstrated FasL and CD63 colocalization in intraluminal vesicles within cytolytic granules in NK cells [100]. Co-localization of ADAM10 in these vesicles was not defined; however, this information would be of interest since the Tspan CD63, along with CD9 and CD81 present in Tspan-rich microdomains, is a paradigmatic marker of exosomal vesicles, which often carry ADAM10 [101,102] (Figure 1).

One of the mechanisms elucidating ADAM10 site-dependent biology is Notch cleavage after the fusion of Golgi-derived vesicles containing ADAM10 with endosomes containing endocytosed dissociated Notch [103]. ADAM10 stimulated by diacylglycerol-PKC can facilitate noncanonical Notch activation induced by TCR in CD4^+^ T-cells. This process has been reported to occur in endosomes after Notch endocytosis, suggesting a conformational relief mechanism acting on Notch autoinhibition in the acidic endosomal milieu [104].

We also demonstrated that in mesenchymal stromal cells (MSC) from Hodgkin lymphoma (HL), ADAM10 binds to specific fluorescent inhibitors targeting its active site, colocalizing with Rab5-positive early endosomes and with LysoTrackerRed in endolysosomal vesicles [105]. This might suggest that accumulation in the endolysosomal compartment of ADAM10 itself occurs together with some substrates, such as TNFα or FasL, in HL stromal and tumor cells upon treatment with ADAM10 inhibitors.

These considerations raise the question of how ADAM10 recognition of substrates stored intracellularly, requiring proper mobilization and ADAM10 co-localization, or clustered in complex, takes place; ADAM10 and ADAM17 can actually be activated upon cell stimulation by exogenous and endogenous stress or danger signals, including inflammatory mediators and small signaling molecules, and their interacting proteins have been reported [13,31,64,65].

Table 1 highlights the fine dynamics of ADAM10 proteolytic activity and its regulation.

## 5. ADAM10 Relocation and Related Function

### 5.1. Redistribution Induced by Regulatory Proteins

Several lines of investigation have identified some of the molecular determinants and signaling proteins that regulate ADAM10 activation as a function of its intracellular distribution [77,99] (Figure 1). This fact has profound implications for cell structure, cell membrane integrity, tissue injury and disparate cell functions related to inflammation and antitumor immune response [10,14,16,34,52,71,79].

Several SH3 domain-containing proteins are involved in endocytosis and cytoskeleton dynamics. The intracellular SH3 binding domain of ADAM proteases is capable of interaction with molecules of the Src family of nonreceptor protein tyrosine kinases (PTK), Grb2, Yes, PI3K p85α subunit and others [19,109]. ADAM10 and ADAM17 SH3 domains have been reported to interact with the SH3 protein SAP97 to change localization [110].

Additional levels of control come from GPCR, responsible for ADAM17 activation and membrane insertion [55]. GPCR and rhomboid-like4 (RHBDL4), for example, communicate to control the extracellular mobilization of bioactive molecules, such as the EGFR ligand transforming growth factor-α, in extracellular vesicles, or in a soluble active form upon shedding by ADAM proteases [111]. Shedding of the ADAM10 substrate HB-EGF, as well, enhances EGFR transactivation by GPCR ligands lysophosphatidic acid and bombesin [112]. The complexity of action and roles in enzyme trafficking and activity rendered GPCR attractive drug targets in diverse pathophysiological contexts [113].

An important discovery for understanding ADAM10 relocation was the dependence on the RabGTPase Rab14 and the Rab GDP-GTP exchange factor (GEFs) FAM116A, all involved in cytoskelatal reorganization. In the absence of these regulators of cell migration and cell-cell junctions, ADAM10 accumulates in a recycling endocytic compartment, blocking its access to the plasma membrane and its shedding activity [106].

### 5.2. ADAM10 Activity in Extracellular Vesicles and in Unexpected Cell Sites

The discovery of transport by extracellular vesicles (EVs) of biomolecules, including a multitude of proteases, opened a new research field aimed at deciphering the fate of these membranous vectors at distance from the site of release, their tissue-specific composition, and the role of cargos in target cells. Members of the Tspan family CD9, CD63 and CD8 are generally found in EVs, where they constitute molecular markers. These molecules might take part to cargo selection, besides vesicle sorting, under control of GTPases, in some instances [114].

Shedding of substrates by ADAM10 and other members of the family has predominantly been described for the cell membrane-associated forms. Actually, for several of these molecules, emerging evidence indicates the existence of processed intra- and extracellular vesicular forms. Processing of the low-affinity IgE receptor CD23, CD44 and L1 by ADAM10 apparently occurs in intracellular endosomes before sorting and release in EVs [74,94,96].

The ADAM10 targets MICA/B have recently been localized intracellularly in normal and tumor tissues [115,116]. These results are in line with previous findings showing MICA/B, in particular MICA008*, shedding and release into intracellular vesicles, before reaching the cell surface and fusing with the plasma membrane [107]. CD30, similarly, has been found in EVs co-expressing ADAM10 [105,108,117]. In these cases, the result of ADAM10 sheddase activity, transported by these vesicles, may lead to alterations of immune reactions. For instance, soluble MICA/B released in the extracellular milieu can impair the function of activating receptors (such as NKG2D) expressed by T or NK lymphocytes, interfering with the response against tumors expressing stress molecules [107,115,116]. In turn, soluble CD30 has been reported to reduce the effect of therapeutic antibodies directed against lymphoma cells expressing CD30 [105,108,117].

ADAM10 itself has been shown to actively take part in exosomal sorting of several of its substrates. Actually, ADAM10 is often considered to bea marker for some classes of EVs, such as ectosomes or shedding vesicles, larger than exosomes (100-1000nm), directly budding from the cell membrane [101,102]. The biogenesis of exosomal metalloproteinases [3] undermines the classical view of static enzymes in defined cell sites, providing the evidence of enzyme relocation and “out of place” activity [118]. Apart from the relevance in basic research, the possible theranostic use of exosomes in cancer, inflammatory and degenerative diseases is a topic of numerous clinical studies [119]. Along this line, we reported that ADAM10 inhibitors localize in exosome-like vesicles (ExoV) in HL cells, where the bioactive form of ADAM10 is also located. ADAM10 sheddase activity carried by these vesicles to bystander lymphoma or stromal cells induces the shedding of TNFα and CD30, with consequent reduction of anti-lymphoma effects exerted by the antibody-drug conjugate brentuximab-Vedotin; this effect is recovered when ExoV contain ADAM10 inhibitors that neutralize the sheddase activity [105].

The intracellular domain of ADAM10, generated upon RIP by ADAM9, ADAM15 and γ-secretase, has been localized to nuclear speckles, an unexpected and remarkable location, with a gene regulatory role, whose functional consequences are yet to be defined [52].

Due to the stress-related regulation of ADAM10, and the fact that stress granules associate with SH3 domain proteins, we are tempted to speculate that ADAM members might be associated with these structures, whose crucial role in cell physiology is an open field of investigation. Moreover, the intrinsically disordered protein regions found in ADAM10 cytoplasmic tail, favoring homodimerization [20], could be another structural clue for exploring its possible association with stress granules. Other proteases, such as the ubiquitin-specific protease USP10, which reduces polyubiquitylation of substrates, have been found in stress granules and in membrane-less organelles [120], possibly representing additional effectors of ADAM10 relocation. The mechanisms fine tuning ADAM10 activation/repression in response to the internal or environmental stimuli await further research.

## 6. Conclusions

The aim of the present work is to contribute a conceptual framework for further studies on ADAM proteases, used as an exemplar of the ways in which enzymes can act in different biological contexts depending on subcellular sites, membrane microdomains and defined stimuli.

It is now clear that ADAM10 does not stand alone waiting for substrates to be cut, crouching at the plasma membrane. ADAM10 works as a protomer (the smallest functional unit), it clusters at specialized membrane microdomains, can sense cell polarity, receives chemical instructions as well as input from signaling networks, and in turn imposes functional constraints to a huge number of effector bioactive molecules. The physical association of enzymes into functional domains conveys and channels information captured from the flux of small molecules and second messengers inside the cell. This scenario is shared by signaling proteases which help cells select actions, decisions and cell fate when acting in transient, dynamic multiprotein complexes at different cell sites, in vesicular, tethered or otherwise aggregated forms, or in the extracellular space. Although substantial evidence of ADAM10 and ADAM17 as cell membrane sheddases represents the pivotal paradigm to understand ADAMs biology, emerging evidence indicates that ADAM10 location and activity are likely under strict regulation by complex and coordinated signaling events involving vesicular transport dynamics and small molecule-regulated signaling. By addressing some accessibility and selectivity issues, these aspects could help sharpening future development of novel theranostic strategies.

To organize functional spatial patterns, the cooperation between proteases, kinases and signaling enzymes implies stabilization by membrane anchoring and physical interactions, in some instances, and discernment of small molecule signaling. The details of how ADAMs, including ADAM10, accomplish all these tasks and the molecular mechanisms by which intracellular ADAMs are activated or stored in an inactive form in vesicular compartments (ER, Golgi, TGN, endolysosomal and extracellular vesicles) remain to be clarified.

## Figures and Tables

**Figure 1 ijms-22-04969-f001:**
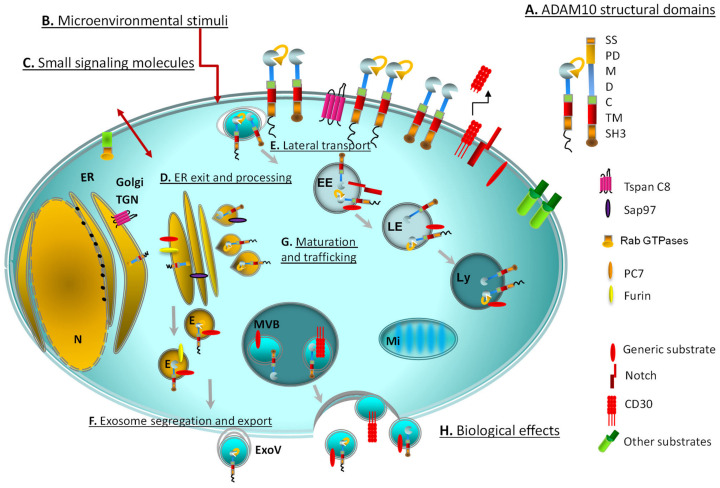
Cellular landscape of ADAM10 processing, maturation and transport showing multiple sites of location and activity. The enzyme is represented with its main structural domains: SS: signal sequence, PD: pro-domain, M: metalloproteinase, D: disintegrin, C: cysteine-rich, TM: transmembrane, SH3: cytoplasmic tail-SH3 binding (**A**). The graphical representation of the cytoplasmic tail is shown as a spherical (ordered) or zigzag (disordered) domain and refers to the role of ADAM10 cytoplasmic tail structure on its activation. Microenvironmental stimuli (**B**) potentially involved in ADAM10 modulation: inflammatory cytokines, such as interleukin-1β or tumor necrosis factor (TNF)α, tumor-derived molecules (soluble, exosomal), endotoxins. Signaling enzymes (RabGTPases) and small molecules (cAMP/cGMP, protein kinase C activators such as diacyl glycerol or phorbol myristate acetate, Ca^2+^ nitric oxide, reactive oxygen species) regulate the intracellular membrane trafficking and association with signaling organelles (**C**). PC7 and furin participate to Golgi processing (**D**). Tetraspannins (Tspans), in addition to Sap97, are involved in the exit of ADAM10 from the ER (**D**) and in the regulation of lateral transport of ADAM10 monomers (**E**), forming enzymatic complexes able to cleave defined substrates (the main molecules are represented and listed). ADAM10 location inside the organelles including exosomes and exosome-like vesicles (**F**) is depicted. Task-driven trafficking (**G**) and association with substrate-specific molecular partners might represent a paradigmatic modality used by a single protein to exert a large number of biological effects (**H**): immune escape and lymphoid or myeloid cell regulation, autoimmunity, allergy/asthma, tumor invasion, cell migration, tissue remodeling, Alzheimer’s and other degenerative diseases. Other substrates: MHC class I chain-related gene A/B (MICA/B), UL16 binding proteins (ULBPs), TNFα, CD95 Ligand, CD23, CD44, L1. E: exosomes; EE: early endosomes; ER: endoplasmic reticulum; ExoV: exosome-like vesicles; LE: late endosomes; Ly: lysosomes; Mi: mitochondria; MVB: multivesicular bodies; N: nucleus; TGN:trans-Golgi network.

**Figure 2 ijms-22-04969-f002:**
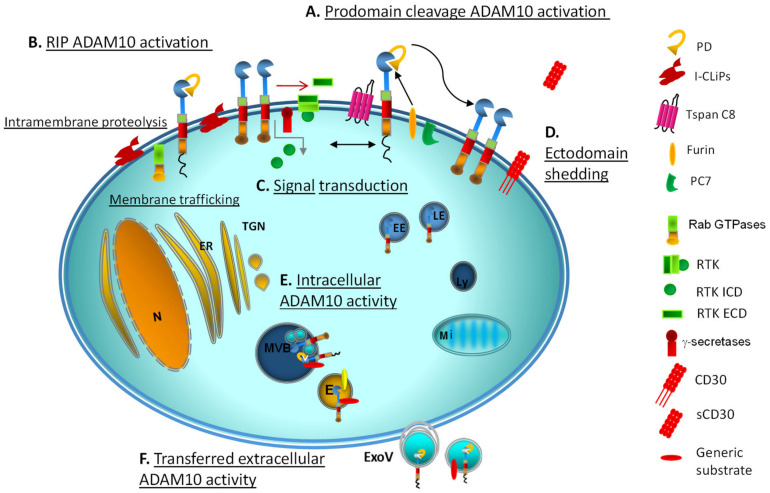
Scheme of the main mechanisms of ADAM10 activation and function. (**A**) Prodomain cleavage takes place by the action of furin and PC7 enzymes; in turn, ADAM10 active site can be autoinhibited by the C-terminal tail of the adjacent protomer, mimicking the substrate. (**B**) RIP engages intramembrane-cleaving proteases (I-CLiPs) through which transmembrane precursors are cleaved to liberate their cytoplasmic or extracellular fragments, enabling them to function at a new location. In turn, ADAM10 participates, for instance, in the cleavage of RTK (**C**) within the extracellular juxtamembrane domain (ECD, red arrow), while γ-secretases provide the cleavage of the intracellular domain (ICD, grey arrow). Another example of ADAM10 sheddase activity is the cleavage of CD30 ectodomain and release as soluble form (**D**). RabGTPase and Tspans are also depicted. Finally, there is evidence for intracellular ADAM10 function, inside the organelles (**E**), and for the extracellular transfer of this enzymatic activity, carried by exosomal like vesicles (**F**). E: exosomes; EE: early endosomes; ER: endoplasmic reticulum; ExoV: exosome-like vesicles; LE: late endosomes; Ly: lysosomes: Mi: mitochondria; MVB: multivesicular bodies; N: nucleus; PD: prodomain; TGN: trans-Golgi network.

**Table 1 ijms-22-04969-t001:** Evidence of ADAM10/17 activity in subcellular sites and multiprotein complexes ^1^.

ADAM10 Target	Site	ADAM10/17 Interacting Proteins/Small Signalling-Accessory Molecules	Docking/Locking/Motor Proteins	Ref.
Notch	cell membrane endocytic/trans-endocytotic vesicles	E3 ubiquitin ligase DTX4, dynamin	dynamin, actin, epsin	[103]
	cell membrane endocytic vesicles	γ-secretase, MT1-MMP, furin		[49]
	endocytic vesicles	TCR, DAG-PKC		[104]
FasL	lysosome, lipid rafts	TCR, Src		[99]
TNFα	tetraspanin web	MEK/ERK	Tspan (CD81,CD9)	[79]
CD23	endosomes	Ca^2+^	clathrin	[96]
CD44	endosomes, exosomes	homodimerization, Ca^2+^		[48,94,97]
L1	endosomes exosomes	Ca++, cholesterol depletion, pervanadate, phorbol esters, Src		[74,94,97]
E-cadherin	adherens junctions	EphA4, EphB2, EphB, GPI-anchored EphB1		[82,91,106]
*S.aureus*α-toxin	adherens junctions	PLEKHA7	Tspan33, PDZD11, afadin	[85]
EphA1	trans-endocytic vesicles	EphA2	clathrin, dynamin,	[84]
EphA5	cell-cell interface	EphA3 (in trans), RTK		[81]
sortilin	endosomes	BDNF, PKC, calmodulin, Ca^2+^		[98]
APP	membrane endosomes	γ-secretase presenilin	Tspan3	[29]
MBP, collagen IV	membrane, microsomes	furin (^209^RKKR cleavage)		[92]
CD30, MICA008/ULPBs	exosomes			[105,107,108]

^1^ Experimental evidence of ADAM10/17 subcellular activity in complex with a variegated repertoire of substrates and cell regulators. Some of the substrates shown are both processed by ADAM10 or ADAM17 in different manners (i.e. sortilin). This distinction was omitted in the table, see references for more detailed information on ADAMs partners and activators.

## Data Availability

Not applicable.

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
