# Peer review of "ADAM10 Site-Dependent Biology: Keeping Control of a Pervasive Protease"

_ijms, 2021, doi:10.3390/ijms22094969_

Round 1
Reviewer 1 Report
In this review, Tosetti et al. discuss the regulation and function of the metalloprotease ADAM10 with a special focus on its location within the cellular environment. The recent observation that ADAM proteases not only mediate shedding events on the cell surface but presumably also cleave some of their substrates in different types of vesicles and compartments is intriguing. To date there are no detailed reviews that focus on the subcellular compartmentalization of ADAMs. While this manuscript is a timely review that summarizes the recent findings in this relevant field some concerns should be considered.
Although the manuscript is overall well-written, it requires moderate editing and proofreading to make the content more accessible to the readers.
In addition, some words and phrases seem to be unusual in the context used. E.g., the word “pervasive” carries a presumption of negativity and is often used with a negative viewpoint. I suggest to change the title to something like “ubiquitous” or “prevalent” instead.
Also, the word choice “foreground” seems to be unusual. The authors presumably mean “focus”?
Other examples are: “sophisticated assemblies” or “unpredicted environments”. Scientific language should be used instead.
More importantly, this manuscript suffers from some structural issues that need to be addressed.
Specifically, chapter #3 reads more like the actual introduction to this review and should be moved accordingly.
Also, regulated intramembrane proteolysis is introduced and discussed in several chapters. This should be reorganized only be introduced in one place. Most commonly it is known in the context of Notch. This aspect should be mentioned in the beginning.
The authors indicate that ADAM10 and ADAM17 share selected substrates. This needs to be more specified. Do the authors mean when overexpressed in vitro, in disease, and/or physiological settings?
Also, this “overlap” might not only be true for Adams 10 and 17 but also for other ADAMs.
Lines 47-56: It is not clear how this paragraph relates to ADAM function.
In general, I wish the authors would provide a more critical discussion of the reviewed content. There are many incidents where findings are merely cited without critical evaluation.
E.g., the authors indicate that RHBDLs can cleave ADAM substrates in the absence of ADAM10 and ADAM17. However, genetic studies suggest that such compensation does not exist, at least during murine development. There are several other statements that need a similar evaluation by the authors. Also, are RHBDL1-4 really considered to be inactive?
The authors need to evaluate the literature in terms of relative importance and put findings into perspective. E.g., while it has been shown that ADAM10 can leave TNF alpha, the major TNF alpha sheddase appears to be ADAM17. Also, iRhoms are critical cofactors for its maturation and should be discussed in this context.
Reviews should help the non-expert readership to distinguish between the “norm” and the “exception”
Overall, this review gives the impression that the catalytic function of ADAM10 on the cell surface might be less defining and emphasizes “niche” observations that are only supported by a limited number of studies. Again, I do think that these studies are important to be discussed and reviewed here but need to be critically evaluated and discussed (e.g., limitations).
Figures
The figures address important points of the ADAM10 biology. However, the vast number of symbols, additionally presented in various forms and colors, currently provides very limited information. These figures could be simplified. Also, does ADAM10 mature on the cell surface?
Minor comments:
Should it be “RIPping” or “RIPing”? Maybe “RIP-dependent”?
“There is also evidence that different TspanC8s and iRhoms are not only responsible for intracellular transport, but also target the ADAMs to distinct substrates.” Citations should be provided.
Organelle abbreviations are missing in the figure legends (figure 2). Instead, they appear in lines 262-264 of the main text.
Consistently either “ADAM10 and -17” or “ADAM10 and ADAM 17” should be used throughout the manuscript.
What is “UPS”? - “unconventional protein secretion”?
Author Response
Reviewer 1
Comments and Suggestions for Authors
In this review, Tosetti et al. discuss the regulation and function of the metalloprotease ADAM10 with a special focus on its location within the cellular environment. The recent observation that ADAM proteases not only mediate shedding events on the cell surface but presumably also cleave some of their substrates in different types of vesicles and compartments is intriguing. To date there are no detailed reviews that focus on the subcellular compartmentalization of ADAMs. While this manuscript is a timely review that summarizes the recent findings in this relevant field some concerns should be considered.
Although the manuscript is overall well-written, it requires moderate editing and proofreading to make the content more accessible to the readers.
We thank the Reviewer both for the overall positive judgement and for the criticisms raised.
In addition, some words and phrases seem to be unusual in the context used. E.g., the word “pervasive” carries a presumption of negativity and is often used with a negative viewpoint. I suggest to change the title to something like “ubiquitous” or “prevalent” instead.
We did not consider “pervasive” as a word indicating a negative characteristic. See also the definition of “pervasive” by the Oxford dictionary: “existing in all parts of a place or thing; spreading gradually to affect all parts of a place or thing.” The word “ubiquitous” is defined as follows: seeming to be everywhere or in several places at the same time. The dynamic concept of “spreading gradually”…is missing. Since in the review we tried to draw attention to the interconnection occurring among ADAM10 localization, trafficking and substrate selection, we would like to maintain the word “pervasive”, if the change is not considered mandatory by the Reviewer.
Also, the word choice “foreground” seems to be unusual. The authors presumably mean “focus”?
The word foreground has been deleted from the text.
Other examples are: “sophisticated assemblies” or “unpredicted environments”. Scientific language should be used instead.
“sophisticated assemblies” has been changed into “highly organized”; “unpredicted environments” into “environments unexpected on the basis of previous knowledge”.
More importantly, this manuscript suffers from some structural issues that need to be addressed.
Specifically, chapter #3 reads more like the actual introduction to this review and should be moved accordingly.
The chapter has been moved and is now chapter # 2.
Also, regulated intramembrane proteolysis is introduced and discussed in several chapters. This should be reorganized only be introduced in one place. Most commonly it is known in the context of Notch. This aspect should be mentioned in the beginning.
The text has been reorganized accordingly. RIP is now treatedin the chapter on its role in ADAM10 activation (the leading actor of the review) with a further description of its role in the context of Notch.
The authors indicate that ADAM10 and ADAM17 share selected substrates. This needs to be more specified. Do the authors mean when overexpressed in vitro, in disease, and/or physiological settings?
Also, this “overlap” might not only be true for Adams 10 and 17 but also for other ADAMs.
We agree with the Reviewer, but the focus of this review is ADAM10.Nevertheless the following sentence has been added to Chapter 2.1, lines 62-66.
“For example, tumor necrosis factor-α (TNF-α) a substrate for both ADAM17 (also named TNF-α convertase or TACE) and ADAM10, is involved in the development of some human tumors where ADAM10 and ADAM17 are overexpressed [7]. On the basis of these data, ADAM10 and ADAM17 have been proposed as biomarker and possible targets of therapy in cancer [8].“
Some major ADAM10 AND ADAM17 shared substrates, namely Notch, TNF-α, EGF, CD44, CX3CL1, IL6R, APP, meprin, L1, Delta, Klotho, as identified in different experimental and/or physiological settings have been listed in paragraph 4.1, lines 371-373.
A selection of relevant literature reviewing the experimental evidence of ADAMs activity in pathophysiological contexts has been cited in this work (i.e. on page 2).
Lines 47-56: It is not clear how this paragraph relates to ADAM function.
No it’s general. Now the introduction has been changed.
This citation was related to the core of our work in that it is an exquisite example of an enzyme with multiple sites of action, and spatial relocation triggered by changes in the metabolic state of the cell.
In general, I wish the authors would provide a more critical discussion of the reviewed content. There are many incidents where findings are merely cited without critical evaluation.
This is partially due to the fact, also noted by the Reviewer, that “…there are no detailed reviews that focus on the subcellular compartmentalization…..” and function of ADAMs…hence the need of a rather structured presentation of evidence on the subject. Nevertheless, we added discussion and considerations in the text.
E.g., the authors indicate that RHBDLs can cleave ADAM substrates in the absence of ADAM10 and ADAM17. However, genetic studies suggest that such compensation does not exist, at least during murine development. There are several other statements that need a similar evaluation by the authors. Also, are RHBDL1-4 really considered to be inactive?
We agree with the reviewer that this point needs clarification. See also lines 216-225.
We did not state that ADAM10 and ADAM17 were absent, we used instead the term inactive.
Actually, in the experimental setting cited (ref 46, Adrain), which took advantage of non-cleavable EGF mutants and shRNA, the broad spectrum metalloproteinase inhibitor batimastat was used to inactivate proteolytic enzymes other than RHBDLs.
We are aware that the genetic evidence of a general role for ADAM10 in EGF secretion, as it occurs in embryonic fibroblasts, is lacking due to the lethality of ADAM10-/- mice.In this experimental context, however, EGF processing and EGFR activation, which follow ADAM10 activity, were exemplar readouts of ADAMs substrates cleavage by RHBDL2.
This point is now synthesized in a more concise phrase focused on a possible crosstalk between ADAMs and active rhomboids in processing RTK ligands, a topic relevant in tumor biology, which catched our attention (lines 222-225).
Regarding the lack of activity of rhomboids in mammals, we referred to catalytically inactive iRhom1 and iRhom2.
The authors need to evaluate the literature in terms of relative importance and put findings into perspective. E.g., while it has been shown that ADAM10 can leave TNF alpha, the major TNF alpha sheddase appears to be ADAM17.This is stated in the text. Also, iRhoms are critical cofactors for its maturation and should be discussed in this context.
These points have been detailed as suggested by the Reviewer:
-specification of the prominent role of ADAM17 in TNFalfa shedding in paragraph 2.1 (lines 62-66)
-role of iRhom2 in ADAM17 maturation in paragraph 2.2 (lines 166-167).
For space and reference limitations the original publication has not been cited (Adrain et al. Science 2012 335(6065):225-8. doi: 10.1126/science.1214400)
Reviews should help the non-expert readership to distinguish between the “norm” and the “exception”
Overall, this review gives the impression that the catalytic function of ADAM10 on the cell surface might be less defining and emphasizes “niche” observations that are only supported by a limited number of studies. Again, I do think that these studies are important to be discussed and reviewed here but need to be critically evaluated and discussed (e.g., limitations).
The statement that the role of ADAM10 and ADAM17 as cell membrane sheddases represents the basis to understand their biological activity has been underlined in the “Conclusion”, lines 587-588.
Figures
The figures address important points of the ADAM10 biology. However, the vast number of symbols, additionally presented in various forms and colors, currently provides very limited information. These figures could be simplified. Also, does ADAM10 mature on the cell surface?
Figures have been simplified, with many particulars described in the legend only.
ADAM10 undergo processing and maturation from the ER to the Golgi network and can be activated (the active form is also called mature, while the inactive protein can be defined immature) on the plasma membrane through the two main mechanisms depicted in Fig.2, but there is increasing evidence that it can be activated and function also in subcellular organelles such as endosomes and exosomes (among others, refs. 73, 74, 94, 96, 104, 105, 112, 117).
Minor comments:
Should it be “RIPping” or “RIPing”? Maybe “RIP-dependent”?
RIPing was proposed as complementary to cleaving…it’s like ….cropping (RIPping) and cutting (cleaving). Anyway the word has been deleted from the paragraph title.
“There is also evidence that different TspanC8s and iRhoms are not only responsible for intracellular transport, but also target the ADAMs to distinct substrates.” Citations should be provided.
Related references have been quoted, including references already present in the original manuscript and with the addition of four new references (namely refs. 34-37 of the revised version of the manuscript)
Organelle abbreviations are missing in the figure legends (figure 2). Instead, they appear in lines 262-264 of the main text.
This is an editing mistake: lanes 262-264 of the previous version are now included in the legend to figure 2.
Consistently either “ADAM10 and -17” or “ADAM10 and ADAM 17” should be used throughout the manuscript.
ADAM10 and ADAM17 is now used in the text.
What is “UPS”? - “unconventional protein secretion”?
Yes, it is. However, UPS abbreviation has been deleted because it does not appear again in the text.

Reviewer 2 Report
Tosetti and collaborators present an interesting review about the ADAM10 endopeptidase, its activation mechanisms, functional activities, regulatory interactions, and the biological consequences during the proteolysis of multi-molecular complexes. The work is well organized and comprehensively written. The field and the general audience will appreciate the topic of the review. I have minor editorial changes and few suggestions that could help to improve the current version of the review.
1) Page 7, the paragraph from lanes 256 to 265 should be part of the figure 2 legend.
2) It is interesting what the authors describe on page 6 ( lanes 228-235) about the autoantibodies against ADAM 10 prodomain in colorectal cancer. In human tumors, which is the status ADAM 10 and Tspan genes (i.e., overexpressed, mutated, deleted)?
3) In section 3.2, it would help to clarify why Tspan cannot work on ADAM17 as well as rhomboid proteins cannot work on ADAM10.
Author Response
Reviewer 2
Tosetti and collaborators present an interesting review about the ADAM10 endopeptidase, its activation mechanisms, functional activities, regulatory interactions, and the biological consequences during the proteolysis of multi-molecular complexes. The work is well organized and comprehensively written. The field and the general audience will appreciate the topic of the review. I have minor editorial changes and few suggestions that could help to improve the current version of the review.
1) Page 7, the paragraph from lanes 256 to 265 should be part of the figure 2 legend.
This is an editing mistake: lanes 262-264 of the previous version are now included in the legend to figure 2.
2) It is interesting what the authors describe on page 6 ( lanes 228-235) about the autoantibodies against ADAM 10 prodomain in colorectal cancer. In human tumors, which is the status ADAM 10 and Tspan genes (i.e., overexpressed, mutated, deleted)?
We agree with the Reviewer that this is an interesting issue but is out of focus of this review that point toward the physiological regulation of ADAM functions. Indeed, to keep clear the message of the review, which is already overcrowded of information, we decided to avoid including pathological implications.
3) In section 3.2, it would help to clarify why Tspan cannot work on ADAM17 as well as rhomboid proteins cannot work on ADAM10.
The following sentence with related references, explaining the specificity of the interaction, have been added in the revised version of the manuscript (lines 171-178).
”These specific binary interactions seem to be regulated by different regions of the molecules. Indeed, TspanC8 and ADAM10 interaction is mediated by their extracellular region, largely involving the cysteine-rich domains (ref.27 Noy), while the interaction of ADAM17 with rhomboids is mainly mediated by its cytoplasmic tail (ref.35 Maretsky). This is not surprising since ADAM10 and ADAM17 share only 30% of amino acids identity in humans. For example, the cysteine-rich domain in ADAM17 is much shorter than in ADAM10 (38 vs. 118 amino acids, respectively), thus it might not be suitable for the interaction with TspanC8.

Round 2
Reviewer 1 Report
The authors have addressed all of my concerns.